# Impact of the COVID-19 Pandemic on the Elective Surgery for Colorectal Cancer: Lessons to Be Learned

**DOI:** 10.3390/medicina58101322

**Published:** 2022-09-21

**Authors:** Catalin Vladut Ionut Feier, Razvan Bardan, Calin Muntean, Andra Olariu, Sorin Olariu

**Affiliations:** 1First Discipline of Surgery, Department X-Surgery, “Victor Babes” University of Medicine and Pharmacy, 2 E. Murgu Sq., 300041 Timisoara, Romania; 2First Surgery Clinic, “Pius Branzeu” Clinical Emergency Hospital, 156 L. Rebreanu Blvd., 300736 Timisoara, Romania; 3Department of Urology, “Victor Babes” University of Medicine and Pharmacy, 2 E. Murgu Sq., 300041 Timisoara, Romania; 4Department of Informatics and Medical Biostatistics, “Victor Babes” University of Medicine and Pharmacy, 2 E. Murgu Sq., 300041 Timisoara, Romania; 5Faculty of Medicine, “Victor Babes” University of Medicine and Pharmacy, 2 E. Murgu Sq., 300041 Timisoara, Romania

**Keywords:** COVID-19 pandemic, colorectal cancer, elective surgery, length of hospital stay

## Abstract

The review investigates the impact of the COVID-19 pandemic on the elective surgical treatment of patients diagnosed with colorectal cancer, and the modifications of the duration of hospital stay scheduled for the surgery. Most of the studies included in our analysis showed a decrease in the number of elective surgical procedures applied to patients with colorectal cancer, varying from 14% to 70% worldwide. We have also observed a significant shortening of the hospital stay in most of the cases, associated with a longer waiting time until hospital admission. In the end, we have performed a synthesis of all the valuable data and advice gathered from real life observations, proposing a strategy to deal with the pandemic and with the large number of cancer patients accumulated during these difficult times.

## 1. Introduction

At the end of 2019, a new strain of coronavirus was detected in China, with symptoms specific for acute respiratory disease, which has spread very quickly in all parts of the world. The virus was named SARS-CoV-2 and the World Health Organization declared it an international public health emergency on 30 January 2020.

The COVID-19 pandemic had a major devastating impact on the functioning of healthcare systems around the globe, many countries allocating all their resources to combat the disease, with more or less success. Many of the cancer patients became reluctant to seek for help, due to infection fears, exposing them to considerable cancer progression risks. Moreover, surgical activity has been severely affected by the spread of the new virus, as most of the hospital activity was dedicated to the isolation and therapy of infected patients. 

As a result, the management of oncological patients received significantly less attention: many cancer patients had their diagnostic visits, staging procedures, and surgical interventions postponed, while some of them had to undergo a different form of therapy, not the one which was indicated initially [1,2]. Furthermore, it is estimated that approximatively 38% of all cancer surgical interventions were canceled worldwide in the first 12 weeks of the pandemic [3]. 

This was mainly due to the reduction of the number of available intensive care beds, which were directed to patients with severe forms of COVID-19 disease, as well as to the intended reduction of patient exposure to the new coronavirus [4]. The patients were instructed to stay home and visit the hospital only in case of more severe symptoms, or in emergency situations. 

The impact of the COVID-19 pandemic was also severe on the diagnosis, treatment and prognosis of patients with colorectal cancer, despite the fact that a 2-month delay for surgery reduces patient survival with >9%, and a 6-month delay with more than 29% [1]. Excepting the cases considered as surgical emergency, many of the interventions were postponed, while some of the patients underwent endoscopic procedures (in case of early tumors) or received neoadjuvant chemotherapy (for advanced stages). As colorectal cancer is associated with low immunity, a poor nutritional status and other significant co-morbidities, it requires additional care and a considerable allocation of medical resources. These patients present an increased risk of SARS-CoV-2 infection and could develop more severe forms of COVID-19 [5,6]. 

Considering all these facts, the aim of our review is to evaluate the impact of the COVID-19 pandemic on the elective surgical treatment for patients diagnosed with colorectal cancer, analyzing the delay of the surgical procedures, and the duration of hospital stay during different periods of the pandemic. 

## 2. Literature Review Methodology

### 2.1. Search Strategy

In order to prepare this narrative review, we have conducted an extensive search, using the three most known databases, PubMed/Medline, Scopus, and Web of Science. The selected time period for the published articles was 2020-2021. To identify the relevant studies, the following key words were used: [“Covid-19”, OR “SARS-CoV-2”, OR “Coronavirus 19 disease”, OR “severe acute respiratory syndrome Coronavirus 2”], AND [“colorectal cancer”, OR “colorectal neoplasm”], AND “elective surgery”. A manual search was also performed in the most significant journals of surgery, to find articles related to our subject. 

### 2.2. Inclusion and Exclusion Criteria

All types of observational studies addressing the impact of the pandemic on the patients who underwent elective surgery published in the English language were considered. Elective surgery refers to surgical interventions that have been indicated in patients with colorectal cancer diagnosed on an outpatient basis, diagnosed by colonoscopy and biopsy. 

### 2.3. Selection Methodology

All retrieved articles were downloaded. After duplicate removal, the eligibility of the studies was verified initially by reading the abstracts. Only the articles which analyzed the impact of the pandemic on the elective surgery for colorectal cancer, including the number of interventions, length of hospital stay, and eventually postoperative mortality were selected.

### 2.4. Data Extraction

We have generated a database with the following information extracted from each study: name of the first author, publication month and year, country (or group of countries), study type (retrospective or prospective), patient sample size, time interval of the evaluation, main findings of the study.

### 2.5. Quality Control

We have utilized the “Adapted Newcastle-Ottawa Quality Assessment Scales” checklist to evaluate the quality of the included studies, which were scored as good, moderate, or poor quality [7].

### 2.6. Selection of Studies 

The database search has gathered a number of 93 articles addressing the topic of elective surgery for colorectal cancer during the COVID-19 pandemic. Nine articles containing duplicated data were removed, leaving 84 articles for the review. We have excluded review articles, case reports, letters to editors and short comments. Subsequently, we have checked if the articles had data regarding the number of elective procedures, comparing the pandemic period with the previous period, and if the length of hospital stay, waiting time for hospital admission, and eventually postoperative mortality were properly analyzed, finding that 23 articles were suitable for our review. The flow of information for the article selection is presented in Figure 1. 

After reading and analyzing the full text versions of the 23 articles, we are presenting below their most important findings. 

## 3. Impact of the COVID-19 Pandemic on the Rate of Elective Surgical Interventions for Colorectal Cancer

Although this review has analyzed data from a large variety of countries, we have observed a similar pattern: the number of elective surgical interventions for colorectal cancer was significantly reduced, especially at the beginning of the COVID-19 outbreak. This was explained by the evolution of the global pandemic, the imposed restrictions, the quarantine in most of the countries, the recommendations limiting/forbidding the visits of the patients to the hospitals, as well as by the fear of cancer patients for coming in contact with the new coronavirus.

In the first paper analyzed by our review, we have looked to the data from a national survey performed in Germany, in April 2020 [8]. A number of 112 surgeons participated in the survey, from 101 hospitals. Most of these hospitals (87%) have reduced their surgical caseload, and a 34% reduction of elective oncological colorectal surgical interventions was recorded.

A UK national study analyzing the administrative hospital data of 14,930 patients with colorectal cancer, undergoing surgery from 1 October 2019 to 31 May 2020, has gathered similar results [9]. The number of elective procedures was reduced significantly, from an average of 386 procedures per week before the COVID-19 pandemic, to an average of 214 procedures per week during the first period of the pandemic. The proportion of laparoscopic procedures was reduced, too, from 62.6% to 35.9%.

In contrast with these findings, a study from the Oxford University Hospital, UK, has compared the first 11 weeks from the initiation of the first national lockdown (on 23 March 2020), with the same time period of 2019 [10]. Being a reference center for colorectal cancer, more cancer resections were operated during the pandemic (53 in 2020 vs. 48 in 2019), many of the patients being taken from more severely affected nearby hospitals. However, more procedures were performed using open surgical techniques (49%), compared with laparoscopic techniques (51%), due to the initial opinion that laparoscopic surgery raises the risk for virus spreading (which was thereafter not validated).

A population-based study, spanning the whole National Health System of England, analyzing all cases of operated colorectal cancer from 1 January 2019 to 31 October 2020, and comparing the situation before and after the COVID-19 pandemic, has found a 31% decrease in the number of surgical interventions, with less laparoscopic procedures and more stoma-forming procedures [11].

Similar results were found in a national survey performed in Ireland, using data from the Hospital Enquiry database, analyzing time intervals of one year, from March to February in 2018–2019, 2019–2020, and 2020–2021 [12]. The survey revealed that during the time interval March–May 2020 there was a 42% reduction of colorectal surgical interventions, and a 30% reduction of elective procedures in 2020, compared with 2018 and 2019.

One of the most COVID-19-affected European countries was Italy, where the rate of elective surgical treatment for colorectal cancer has decreased by 34%, compared to the pre-pandemic period [13]. This was due to the severe epidemiological situation that this country went through, especially at the beginning of the pandemic in March 2020, with over 650 deaths/day being constantly reported. 

A national survey performed in Italy has collected answers from 43 Italian surgical centers with experience in colorectal surgery, during the first pandemic wave, at the beginning of 2019. It concluded that from an average of 12.8 resections/month/center in January 2020, the activity decreased to 8.8 resections during the first wave of the COVID-19 pandemic [14].

In Spain, another country with a high number of patients during the first wave of the pandemic, a national survey, performed from February to April 2020 in 67 colorectal surgery units, has revealed that 32.8% of hospitals stopped elective surgery, and 46.3% of the units reduced their activity [15]. An increased number of patients had to wait for a longer time for elective surgery, but no data were available on the waiting time.

A decrease in the number of surgical interventions was also reported in Poland, with a 51% decrease of colorectal procedures, compared to a similar 7-month period before the pandemic, in 2019 [16].

We have included in this analysis our own study, performed in a Department of Surgery from a large teaching hospital in Timisoara, Romania: we have looked for the number of elective surgical interventions for colon cancer and we have found out that during the first year of the pandemic the number of such interventions has decreased with 42%, compared with the previous two years [17]. 

The first country affected by the COVID-19 pandemic was China, and a study performed in the Department of Colorectal Surgery of the Fudan University, in Shanghai, China, over a 4-month period, at the beginning of the pandemic has shown that there was a 35% decrease in the number of patients undergoing elective surgery [18]. In a large hospital from Beijing, the Department of General Surgery evaluated 4 months (February to May) of 2018, 2019, 2020, and found out that the number of elective interventions was reduced by 34% during the pandemic [1].

Another series studied the activity of a large hospital in Beijing, China. During 1 month before the declaration of the outbreak, there were 95 patients operated on for colorectal cancer, while in the following two COVID-19 months there were 71 patients operated on, less than 36 patients/month [19].

The same situation was reported in India, where there was a decrease of 40% in curative interventions and an 80% decrease in palliative interventions [20]. From around 90 patients operated on per month in a specialized Mumbai hospital before the pandemic, the figures have decreased to 50 patients/month during the first pandemic wave.

The effect of the pandemic could be felt even in countries with more powerful healthcare systems, as Austria: in the Academic Hospital Feldkirch the number of elective colorectal surgical interventions fell initially by 71.4%, due to both restrictions imposed by the authorities and to patients’ fear of becoming infected with the new coronavirus [21]. Moreover, the number of patients presenting in T_4_ stages has increased significantly during the COVID-19 pandemic. The authors note that after the implementation of measures designed to minimize the impact of the pandemic, in the second half of 2020 the number of elective procedures has increased with 17.24% (compared with the first half of 2020), but it was significantly lower than in the previous year.

Even an important study by the Department of Surgery from United States has reported similar figures: during the initial wave of the COVID-19 pandemic, the average surgical case rate for elective colorectal procedures has decreased to 2.4 patients per week (from an average of 9.2 patients per week in the previous two years), recording a 74% relative decrease [22].

From South America, we have found a study performed in the Department of Surgery of the San Paolo University Hospital in Brazil, over a 6-month period (March–September 2020). The main finding was that the number of elective surgical interventions for colorectal cancer was significantly reduced in the favor of emergency procedures [23].

Despite the COVID-19 elimination strategy pursued by New Zealand, which isolated all positive patients and contacts and blocked international travel, there was a significant reduction in the number of elective colorectal resection procedures during March and April 2020; counting the whole year of 2020, the reduction of elective surgery was less marked (there was only a 1% reduction, compared with the previous year) [24].

All these findings were confirmed by an international survey carried out in 84 countries, analyzing the answers of 1051 surgeons operating on colorectal cancer, which showed that during the first phase of the pandemic wave 70.9% of cases diagnosed with colon cancer were postponed [25]. In 58.3% of patients who underwent elective colorectal surgery, the delay was caused by changes in the surgical schedule of hospitals, which had to adapt to the reduced number of patients in the patients’ room, to new epidemiological triage rules, and to the reduced availability of the medical staff. Moreover, 26.3% of the patients scheduled for elective surgery finally had to be operated on as an emergency, due to disease progression and complications.

In another large international study published by the CovidSurg Collaborative Group, the estimated global cancellation rate of elective colorectal surgical interventions was 37.7%, over a 12-week period, at the peak of the first wave of the COVID-19 pandemic [3].

The main findings of the above-mentioned studies can be found in Table 1. 

## 4. Impact of the COVID-19 Pandemic on the Duration of Hospital Stay

Another important parameter of our analysis was the average length of hospitalization of the patients diagnosed with colorectal cancer who underwent elective surgery during the pandemic. Most of the studied articles have concluded that the average hospitalization time has decreased significantly during the COVID-19 pandemic, while one article has shown a longer duration of hospitalization. 

During the initial phase of the pandemic, the average length of hospital stay was 4 days in the Oxford University Hospital, while in 2019 the average hospital stay was 5 days [10]. A similar study conducted at Queens Hospital Burton, in England, showed an average hospital stay of 5 days during the pandemic, compared to 8 days in the previous period [26]. The main motivation for this reduction was the need for exposure reduction to SARS-CoV-2 during hospital admission.

During a study performed in a specialized center in Torino, Italy, patients had an average postoperative hospital stay of 5 days during the pandemic, which was comparable with the postoperative stay before the pandemic, also of 5 days [13]. 

In a study conducted in a tertiary hospital in Sao Paolo, the average length of hospitalization of patients was 11.7 days during the pandemic [23]. The same average value was obtained in a study conducted in Madrid at a hospital specialized in the care of oncological patients [27].

In a large Beijing hospital, the average length of postoperative hospital stay was 9.6 days during the pandemic compared to 12.1 days in 2018 (*p* = 0.015 between the two periods) [1]. However, the average total hospital stay was 18.5 days in 2020, compared to 21.3 days in 2018.

Another study performed in the Department of Surgery of the Fudan University, in Shanghai, China, over a 4-month period, at the beginning of the pandemic, has shown that total hospital stay was longer during the pandemic (13.2 days on average), compared with the previous year (11 days on average) [18]. Although this study contradicts all the previously presented articles, which have shown that the patients spent less time in the hospital during the pandemic, it has some legitimate justifications. In the center from Shanghai, China, the hospitalized patients had to wait for 3 days before surgery in isolation, after initial SARS-CoV-2 RT-PCR testing, under closed medical observation, in order to detect any suspicious signs of COVID-19 disease. 

Finally, in a large study conducted on 2073 cases in 270 hospitals from 40 countries, the average length of hospital stay decreased from 7 days (minimum 5 days–maximum 10 days) before the pandemic, to 6 days during the pandemic (minimum 4 days–maximum 8 days, *p* < 0.001) [28]. The attending physicians wanted to reduce the contamination risk by shortening as much as possible the contact with the medical staff and the hospital environment; where possible, the patients were distributed in a smaller number in hospital rooms than in the pre-pandemic period. Furthermore, after the surgery, if the patients had intestinal transit, were hemodynamically stable, and had no postoperative complications, they were discharged (Table 2).

## 5. Waiting Time for Hospital Admission

During the COVID-19 pandemic, an extended waiting time for hospital admission was recorded. In Beijing, China, the waiting time increased from 7 days to 13.5 days in one department, and from 7.95 days to 9.59 days in another department of surgery [1,19]. 

In the Queens Hospital Burton, England, the time interval from the multidisciplinary meeting to hospital admission has increased from 23 to 29 days on average [26].

An international survey analyzing data from 84 countries has concluded that the delay from diagnosis to surgery varied from 5 to 8 weeks [25]. 

Most of the 112 colorectal surgeons participating to a survey in Germany have agreed that an acceptable delay for elective surgery should not be more than 2 weeks [8]. These data were confirmed by several other articles, which concluded that 2 weeks of extended waiting time before surgery are acceptable [10]. 

## 6. Impact on Postoperative Mortality

The available data on the 30-day postoperative mortality were significantly more limited, with just a few of the reviewed articles presenting such an important outcome parameter. During the CovidSurg study, which included 2073 patients electively operated on for colorectal cancer, the recorded 30-day postoperative mortality was 1.8% in 2020, compared with 1.1% in the years before [28]. We should note that this mortality figure did not include patients with COVID-19 postoperative infection. Significant independent predictors were: anastomotic leak, male gender, age over 70 years, stage IV, total/subtotal panproctocolectomy.

Similar figures were reported in other studies: a national survey performed in England has recorded a 30-day postoperative mortality increase from 0.9% in 2019 to 1.2% in 2020 [9]. The study from Sao Paolo, Brazil has reported a 2.1% mortality in non-COVID-19 patients during the pandemic, while a specialized center in India has reported an increase from 1% in 2019 to 1.2% in 2020 [23,29]. 

## 7. Discussion

The SARS-CoV-2 pandemic has had a global impact on health systems everywhere. In the first phase, all efforts were directed towards the diagnosis and treatment of patients infected with the new coronavirus, while patients diagnosed with cancer were deferred. As a consequence, there has been a significant decrease in the number of elective surgical interventions performed for the treatment of colon cancer on all continents during the pandemic. 

The average length of hospitalization also decreased, with multiple studies showing a significant difference between the pandemic period and the previous period, as surgeons preferred to shorten the hospitalization of patients, for reasons such as reducing the risk of exposure with the new coronavirus, as well as due to patients’ fear of prolonged contact with the medical system. 

Considering all these factors, we expect that in the next period of time the number of colon cancer patients requiring surgery will increase significantly. Moreover, these patients are expected to show more advanced stages of disease, with a higher risk for developing intestinal occlusion and other complications that require emergency surgery, finally increasing the mortality risk. Taking into account the fact that we already experience an increased workload for benign disease, we can count on a long-term negative impact on our surgical activity.

One of the most significant limitations of our review was the fact that the vast majority of the articles have analyzed only the most dramatic and intense period of the COVID-19 pandemic, when the disruptions were the most severe, both at the level of the society and of the medical systems. It would have been very useful to assess the impact of all the adaptation strategies on the surgical activity and on patient outcomes, but at least at this timepoint there are only a few published articles covering this topic. Another important limitation was the lack of good quality data in most of the reviewed articles—the authors have compared the pre-pandemic and pandemic periods in terms of number of surgical interventions but did not provide the total number of patients which were postponed; the availability of this data could have allowed us to make odds ratio calculations, comparing the results of different institutions. 

## 8. Learning Points

To conclude on a more positive note, we have tried to synthetize some of the ideas and strategies which have emerged from our review of articles, as a guide for a future pandemic crisis. 

Diagnostic endoscopy and imaging for staging purposes should have the highest priority, to allow a quick diagnosis and therapeutic decision [8,10,29]. Scoring systems for patient prioritization should be implemented, such as MeNTS [30]. Patients who completed neoadjuvant therapy, patients with aggressive forms of cancer, and those with acute symptoms should have priority for elective surgery [31]. 

Routine testing of all admitted patients and hospital staff is needed in order to reduce patient exposure; additionally, staff vaccination significantly reduces the infection risks in the hospital and should be strongly recommended [1,20,29]. Routine use of a preoperative nasopharyngeal swab testing was clearly associated with a lower rate of postoperative pulmonary complications, preventing pre-symptomatic patients from eventually developing severe COVID-19 disease after surgery, and reducing the cross-infection risk [32]. 

Staff allocation should allow for a normal activity in the surgery clinic, operating theatre, and intensive care station. The planning of the elective procedures should be coordinated with the availability of all services in the hospital, especially with the number of available beds in the intensive care departments [20,29,33]. Multidisciplinary meetings should be organized online, to reduce physician exposure—this measure has proven its value in most of the situations during the COVID-19 pandemic [10,26]. 

The interactions with the patients should be reduced as much as possible, especially at the peak of the pandemic events: telemedicine should be routinely used for the preoperative evaluation (including new patient referrals who need triage before investigations) and postoperative follow-up [10,18,21]. Additionally, non-local patients should undergo follow-up tests at their local family physicians or hospitals, reducing their exposure caused by travelling [20]. If possible, the patients should isolate themselves at home for at least 14 days before hospital admission, to further minimize their exposure to the virus [31]. 

In a limited number of cases, the surgery should be delayed for a few weeks—this is the case especially in smaller hospitals, which do not have the possibility to send their patients to larger specialized centers. In order to reduce the impact on patient survival in these cases, clinical trials should be designed to confirm the benefit of neoadjuvant oral chemotherapy [18,29,34].

There is a need for clear guidelines for virus spread prevention in the surgical departments, and especially in the operating theaters, as not all countries have good air filtration systems with negative pressure rooms. The HEPA filters installed in the air conditioning systems and smoke aspirators are very important for virus spread reduction [14]. During patient intubation, only the anesthesia personnel should be allowed in the operating room [33]. Leaking of CO_2_ in laparoscopy should be reduced as much as possible, by changing leaking ports, making smaller skin incisions, using balloon trocars, and air evacuation systems (AirSeal^®^); moreover, a maximum intraabdominal pressure of 8–10 mm Hg should be maintained in laparoscopic and robot-assisted surgery [13]. Surgical techniques should be also revised, if possible: careful sharp dissection with mechanical hemostasis should be favored, reducing the generation of smoke by cauterization [24]. In larger hospitals, it is recommended that there should be an operating theatre dedicated just for COVID-19 patients, minimizing the contamination risk for the patients undergoing elective surgery [35].

The duration of postoperative intensive care and overall hospitalization time should be shortened, to reduce patient exposure, and to give other patients the opportunity to be operated on [1,18,20,23,29]. Appropriate diagnostic measures should be taken as early as possible to differentiate COVID-19-related fever from postoperative fever, including repeat swab testing and chest CT [31,36]. An eventual transfer of patients to satellite surgery teams with less activity should be encouraged, if they are able to provide good quality surgery and comparable postoperative care [3,12,15].

In order to reduce the negative impact on the surgical training of the residents, we should implement strategies to ensure adequate learning of the procedures. In many cases, the use of e-learning, virtual reality computer simulators and pelvitrainers for laparoscopic/robot-assisted surgery could be very useful [21,29,37]. Moreover, rotations to departments with a higher surgical volume are recommended, if possible [29].

## Figures and Tables

**Figure 1 medicina-58-01322-f001:**
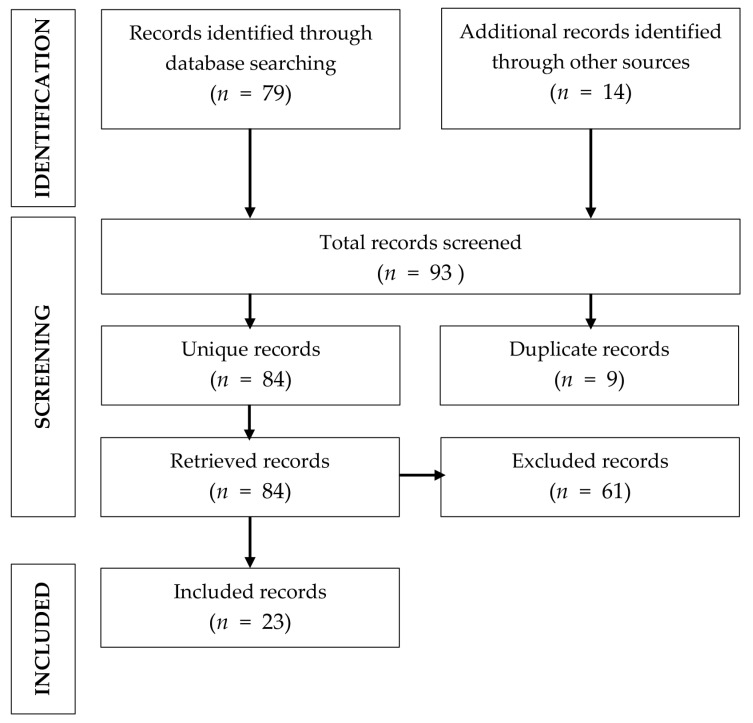
Flow diagram of the review phases.

**Table 1 medicina-58-01322-t001:** Studies analyzing the impact of the COVID-19 pandemic on the rate of elective colorectal surgery.

Study Authors	Center, Country	Analyzed Period	No. of Patients	No. of Surgeons	No. of Centers	Elective Surgery Decrease
Brunner et al. [8]	Germany	April 2020	12,423	112	101	34%
Kuryba et al. [9]	England, UK	1 October 2019–31 May 2020	3227	-	146	26.6%
Merchant et al. [10]	Oxford University Hospitals, UK	23 March 2020–7 June 2020	85	-	1	52.51%
Morris et al. [11]	NHS England	April 2020	1378	-	National database	31%
Donlon et al. [12]	Hospital InpatientEnquiry Database, Ireland	1 March 2020–28 February 2021	1072	-	National database	30%
Allaix et al. [13]	University of Torino, Italy	9 March 2020–15 April 2020	74	-	1	34%
Caricato et al. [14]	Italian ColoRectal Anasto-motic Leakage (iCral) Study Group, Italy	1 January 2020–27 March 2020	1328	-	43	30.37%
de la Portilla et al. [15]	National Survey of Colorectal Surgery Units, Spain	February 2020–April 2020	-	-	67	-
Koczkodai et al. [16]	National Institute of Oncology, Warsaw, Poland	January 2020–July 2020	-	-	-	51%
Feier et al. [17]	1st Department of Surgery, Timisoara, Romania	26 February 2020–1 October 2021	147	-	1	42%
Xu et al. [18]	Fudan University, Shanghai, China	1 January 2020–3 May 2020	710	-	1	14.25%
Cui et al. [1]	Department of General Surgery, National Center of Gerontology, Beijing, China	1 February 2020–31 May 2020	67	-	1	34%
He et al. [19]	Chinese Army General Hospital, Beijing, China	20 January 2020–20 March 2020	71	-	1	63%
Kumar et al. [20]	Tata Memorial Centre, Mumbai, India	Jane 2020–May 2020	257	-	1	39.88%
Tschann et al., [21]	Academic Hospital Feldkirch, Austria	1 January 2020–31 December 2020	63	-	1	11.26%
Yoon et al. [22]	Los Angeles, USA	16 March 2020–23 April 2020	13	-	1	74%
Sobrado et al. [23]	Sao Paulo, Brazil	10 March 2020–9 September 2020	103	-	1	-
Gurney et al. [24]	New Zealand	January 2020–August 2020	-	-	20	1%
Santoro et al. [25]	Worldwide (84 countries)	20 May 2020–10 June 2020	-	-	1051	58.3%
COVIDSurg [3]	Worldwide (71 countries)	12 weeks in 2020	-	-	359	37.7%

**Table 2 medicina-58-01322-t002:** Studies analyzing the impact of the COVID-19 pandemic on the duration of total hospital stay.

Study Authors	Center, Country	Analyzed Period	No. of Patients	No. of Centers	Hospital Stay Pre-Pandemic (days)	Hospital Stay during Pandemic (days)
Merchant et al. [10]	Oxford University Hospitals, UK	23 March 2020–7 June 2020	85	1	5	4
Rashid et al. [26]	University Hospitals of Derby and Burton, England, UK	1 March 2020–30 April 2020	22	1	8 ± 9	5 ± 2
Allaix et al. [13]	University of Torino, Italy	9 March 2020–15 April 2020	74	1	5	5
Sobrado et al. [23]	Sao Paulo, Brazil	10 March 2020–9 September 2020	103	1		11.7 ± 9.3
Cui et al. [1]	National Center of Gerontology, Beijing, China	1 February 2020–31 May 2020	67	-	21.3	18.5
Xu et al. [18]	Fudan University, Shanghai, China	1 January 2020–3 May 2020	710	1	11	13.2
COVIDSurg [28]	Worldwide (40 countries)	First COVID case to 19 April 2020	2073	270 hospitals	7	6

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
