# Peer review of "Impact of the COVID-19 Pandemic on the Elective Surgery for Colorectal Cancer: Lessons to Be Learned"

_medicina, 2022, doi:10.3390/medicina58101322_

Round 1
Reviewer 1 Report
The subject of the review is very interesting, and of interest for the readers. The authors perform a systematic review of the impact of Covid19 pandemic on the elective surgery for colorectal cancer. The article is well structured and written in a clear and concise manner.
However, there are some issues:
1. Most of the studies included in the review (tabel 1) are comunicating data for the first wave of the covid-19 pandemic ( march-june 2020), which correspond to a strict lockdown in most countries. There were significant efforts to priotise the access of the oncological patients to therapy in the following months. As the pandemic lasts for more than 2 years, a dynamic of this process should be included in the analysis.
2. A paragraph with the limitation of the study should be addded in the discusssions.
4. Prisma checklist and Amstar grading of the quality of the studies included in the review should be addded as supplementary files.
5. Conclusions should be re-written as paragraph ( not bulleted) and should reflect only the data analysed in the article.
As minor issues:
Line 170: Should it be March to May instead of March to March?
Table 1: for refference 10, the period of study should be 2020 instead of 2019?
Reviewer 2 Report
The Authors should be complimented for this review on the impact of Covid-19 pandemic on elective colorectal cancer surgery. The manuscript is clear and well written. The final learning points are of great value.
However, few corrections are needed before considering final publication:
- This study is structured as a systematic review according to PRISMA guidelines. However, the structure of the manuscript is more similar to a narrative review. I believe the Authors should adjust the format accordingly.
- The study misses the quality control analysis which is described in the methods. A table and a section should be added evaluating this important aspect.
- Line 162-163: are the percentages correct or should they be swapped between open and laparoscopic?
- Line 242: % is written twice
- Table 2: The data in the column "hospital stay" should be highlighted if they are pre-operative or total hospital stay as described in the text in order to be more precise.
Round 2
Reviewer 1 Report
The authors responded to all queries and revised the manuscript accordingly.
I have no further issues.
Author Response
Thank you for your consideration. We have tried to further improve the manuscript.